# Targeting Mitochondrial IDH2 Enhances Antitumor Activity of Cisplatin in Lung Cancer via ROS-Mediated Mechanism

**DOI:** 10.3390/biomedicines11020475

**Published:** 2023-02-07

**Authors:** He Li, Jiang-jiang Li, Wenhua Lu, Jing Yang, Yunfei Xia, Peng Huang

**Affiliations:** 1State Key Laboratory of Oncology in South China, Collaborative Innovation Center for Cancer Medicine, Sun Yat-sen University Cancer Center, 651 Dongfeng East Road, Guangzhou 510060, China; 2Metabolic Innovation Center, Sun Yat-sen University, Guangzhou 510060, China

**Keywords:** isocitrate dehydrogenase 2, cisplatin, lung cancer, cell death, reactive oxygen species, redox metabolism

## Abstract

Mitochondrial isocitrate dehydrogenase 2 (IDH2) is an important metabolic enzyme in the tricarboxylic acid cycle (TCA) cycle. Our previous study showed that high expression of wild-type IDH2 promotes the proliferation of lung cancer cells. This study aims to test the potential of targeting IDH2 as a therapeutic strategy to inhibit lung cancer in vitro and in vivo. First, we analyzed the available data from the databases gene expression omnibus (GEO) database to evaluate the clinical relevance of IDH2 expression in affecting lung cancer patient survival. We then generated a stable IDH2-knockdown lung cancer cell line using a lentivirus-based method for in vitro and in vivo study. Cell growth, apoptosis, cell viability, and colony formation assays were conducted to test the sensitivity of lung cancer cells with different IDH2 expression status to cisplatin or radiation treatment in vitro. For mechanistic study, Cellular oxygen consumption and extracellular acidification rates were measured using a Seahorse metabolic analyzer, and reactive oxygen species (ROS) generation was analyzed using flow cytometry. An animal study using a xenograft tumor model was performed to further evaluate the in vivo therapeutic effect on tumor growth. We found that high IDH2 expression was associated with poor survival in lung cancer patients undergoing chemotherapy. Inhibition of IDH2 significantly enhanced the anticancer activity of cisplatin and also increased the effect of radiation against lung cancer cells. IDH2 was upregulated in cisplatin-resistant lung cancer cells, which could be sensitized by targeted inhibition of IDH2. Mechanistic study showed that abrogation of IDH2 caused only minimal changes in oxygen consumption rate (OCR) and extracellular acidification rate (ECAR) in lung cancer cells, but induced a significant increase in ROS, which rendered the cancer cells more sensitive to cisplatin. Pretreatment of lung cancer cells with the ROS scavenger N-acetyl-cysteine could partially rescue cells from the cytotoxic effect of cisplatin and IDH2 inhibition. Importantly, abrogation of IDH2 significantly increased the sensitivity of lung cancer cells to cisplatin in vivo.

## 1. Introduction

Lung cancer is the leading cause of cancer-related death worldwide [1]. Non-small cell lung cancer (NSCLC) accounts for about 85% of all lung cancer cases, and includes lung adenocarcinoma (ADC), squamous cell lung carcinoma (SCC), and large cell lung carcinoma (LCC) [2]. Although multiple therapeutic options, including targeted therapies and immunotherapy, are available for the treatment of lung cancer, and despite advances in cancer treatment including immunotherapy, platinum-based chemotherapy is still a commonly used treatment option [3,4,5]. Unfortunately, the overall survival (OS) of NSCLC patients remains unsatisfactory, with a 5-year OS rate of approximately 19% [6]. Certain subtypes of NSCLC, particularly lung ADC, are often resistant to conventional chemotherapies [7]. Therefore, understanding the drug-resistant mechanisms and development of new effective treatments are important tasks required to improve the outcomes among patients with lung cancer.

Although the presence of reactive oxygen species (ROS) can facilitate tumor growth/progression and promote certain drug-resistant phenotypes, high levels of ROS are detrimental to cells [8]. It is known that antioxidant capacity is elevated in some cancer cells and that it confers drug resistance [8,9]. Indeed, reprogramming of energy metabolism is now recognized as a hallmark of cancer [10]. In particular, cancer cells can increase the uptake of glucose and glutamine for nicotinamide adenine dinucleotide phosphate (NADPH) and glutathione (GSH) production to balance intracellular ROS [11,12]. Therefore, disabling NADPH and GSH production has been considered a potential strategy to sensitize tumor cells to chemotherapy and other antitumor treatments [13,14].

The wild-type isocitrate dehydrogenase 2 (IDH2) is a mitochondrial enzyme that catalyzes the interconverts between isocitrate and α-ketoglutaric acid (α-KG) in the TCA cycle, using NADP+ and NADPH for the redox reaction. Thus, abrogation of wild-type IDH2 could cause significant alterations in mitochondrial metabolism and redox status [15,16]. Furthermore, gain-of-function mutations in IDH1 and IDH2 have been discovered in several cancers, and IDH inhibitors, including ivosidenib (AG-120) and enasidenib (AG-221), are now clinically available for treatment of cancers with IDH mutations [17]. In addition to the mutated versions, wild-type IDH2 has been shown to play a role in antioxidant defense [18,19] and to promote cell growth and survival of cancer cells [20,21,22]. In a previous work, we found that IDH2 was upregulated in lung cancer and promoted lung cancer cell proliferation and tumor growth [23]. However, the mechanism by which IDH2 promotes lung cancer and its therapeutic potential have yet to be further investigated. In this study, we examined the association between IDH2 and survival among lung cancer patients undergoing cisplatin chemotherapy and radiotherapy. We also examined the effects of combining IDH2 inhibition with cisplatin or radiation for the treatment of lung cancer and explored the mechanisms underlying the combination effect, with a focus on the impact of IDH2 abrogation on mitochondrial metabolism and ROS generation.

## 2. Materials and Methods

The ReagentsAnnexin V-FITC and propidium iodide (PI) were obtained from BD Biosciences (San Jose, CA, USA). The IDH2 inhibitor AGI-6780 was from Selleckchem (Houston, TX, USA). The antibodies used to detect IDH2 were from Abcam (Cambridge, UK, ab55271), and the antibody used to detect β-actin was from Cell Signaling Technology (Danvers, MA, USA, cst4970).

Cell culture

A549 and HCC827 cell lines were cultured in DMEM with 10% FBS; A549-CR cell line was purchased from Cell Resource Center, Peking Union Medical College (Beijing, China), and cultured in McCoy’s 5A with 10% FBS, as recommended by the vendor. All cell lines were incubated in an incubator at 37 °C with 5% CO_2_. For comparison of CDDP cytotoxicity, both cell lines were first cultured inDMEM+10% FBS for several days before being subjected to drug exposure and cell viability assay.

Generation of IDH2-overexpressing and IDH2-knockdown cells

The shRNA plasmids (GV-248-sh-IDH2#1 and GV-248-sh-IDH2#2) and the non-target control vector GV-248 were purchased from Genechem (Shanghai, China). For lentivirus production, each plasmid was co-transfected with the packaging (psPAX2) and envelope (pMD2.G) vectors into HEK293T cells. Lentivirus was harvested at 48 h post-transfection from the supernatants. A549 and HCC827 cells were infected with the lentivirus and selected in 1 μg/mL puromycin (Selleckchem, China) for three days.

Assays of cell proliferation and cell viability

For cell proliferation analysis, cells were seeded into a 6-well plate (8 × 10^4^ cells/well), and cell numbers were counted using the trypan blue exclusion method in an auto-counting chamber. Apoptosis was stained with an annexin V-FITC/PI kit and analyzed using a Beckman cytoFLEX flow cytometer (Beckman Coulter Life Sciences, Indianapolis, IN, USA). Cell viability was determined in 96-well plates (2 × 10^3^ cells/well) using the MTS reagent from Promega (Madison, WI, USA). The MTS regent was added into culture medium with a ratio of 20:100 and incubated at 37 °C with 5% CO_2_ for 3 h. The culture medium was used as the background control. The absorbance value at 490 nm was measured using a MultiSkan plate reader (Thermo, Helsinki, Finland). For colony formation assay, cells were seeded into 6-well plates (2 × 10^3^ cells/well) and cultured with CDDP (dissolved in warm water) for two weeks. The culture medium was changed at day 7; cell colonies were stained with crystal violet and counted. We also determined the impact of an IDH2 inhibitor (AGI-6780) and cisplatin by incubating lung cancer cells with various concentrations of the compounds and performing the above assays.

Radiation assay

To determine the effect of AGI-6780 and radiation on colony formation, cells were seeded into 6-well plates (1 × 10^3^ cells/well) and cultured for 24 h, treated with IDH2 inhibitor (AGI-6780) for four hours, and then exposed to the indictated doses of radiation. Cell colonies were counted after two weeks.

RT-qPCR

Total RNA was isolated using the Trizol reagent (Thermo Fisher Scientific, Waltham, MA, USA) and then converted to cDNA using a reverse transcription kit (Thermo Fisher Scientific, Waltham, MA, USA). Quantitative real-time polymerase chain reaction (qRT-PCR) was performed using the ABI7500 qPCR system (Applied Biosystems, Waltham, MA, USA). The re-actions contained the indicated cDNA, primers, and SYBR^®®^Green Real-Time PCR Master Mixes (Thermo Fisher Scientific). The sequences of primers were as follows: beta-actin forward AGAGCTACGAGCTGCCTGAC, reverse AGCACTGTGTTGGCGTACAG; IDH2 forward CCTGCTCGTTCGCTCTCCA, reverse ACGGGTCATCTCATCACCATC.

Western blot analysis

Cells were harvested and washed twice with cold PBS, lysed in protein-IP lysis buffer with protein inhibitors, and centrifuged at 12,000× *g* for 20 min. The supernatant containing protein lysates (20 μg, as quantified by the BCA Protein Assay Kit, Thermo Fisher Scientific) was subjected to SDS–PAGE and transferred to a PVDF membrane (Millipore, Bedford, MA, USA). Membranes were blocked with 5% non-fat milk for 1 h at room temperature and then incubated with the specific primary antibody, washed with TBST, and then incubated with the corresponding HRP-conjugated secondary antibody. Protein bands were visualized using the Western lightening plus-ECL kit (Thermo Fisher Scientific).

ROS assay

Cells (5 × 10^4^) were seeded into a 6-well plate and incubated with AGI-6780 and or cisplatin for 24 h, followed by staining with Dihydroethidium (1 μM). Cells were then harvested, washed twice with PBS, and analyzed using the Beckman cytoFLEX flow cytometer.

Analysis of cellular and mitochondrial metabolism

Cells (5 × 10^4^) were seeded into a 24-well plate and incubated with AGI-6780 and or cisplatin for 24 h. The oxygen consumption rate (OCR) and extracellular acidification rate (ECAR) were measured using a Seahorse XF24 metabolic analyzer according to the proto-cols recommended by the manufacturer (Agilent Technologies, Inc., Santa Clara, CA, USA). Oligomycin (1 μM), FCCP (1 μM) and antimycin A + rotenone (0.5 μM) was used in OCR analysis as indicated. Exogenous glucose (10 mM), oligomycin (1 μM), and 2-DG (50 mM) were used in the ECAR measurement, as indicated in the respective figure legends. Mitochondrial metabolic parameters, including basal respiration, mitochondrial ATP production, maximal respiration capacity, spared respiratory capacity, and proton leak were calculated from the OCR curves according to the methods provided by the manufacturer of the Seahorse metabolic analyzer (Agilent Technologies, Inc.).

Animal study

Animal experiments were conducted according to protocols approved by the Sun Yat-sen University Animal Care Committee. Nude mice (6 weeks old) were injected in their flanks with either A549 cells with IDH2-knockdown cells (2 × 10^6^) or the corresponding control cells. Tumor growth was assessed by measuring the length and width of the tumors. Tumor size was calculated using the following formula: length × width^2^/2. For drug treatment, cisplatin was injected intraperitoneally at a dose of 3 mg/kg/week.

Analysis of databases

Publicly available datasets were from Kaplan–Meier Plotter database (http://kmplot.com/analysis/index.php?p=service&cancer=lung, accessed on 1 November 2020) and GEO databases, as indicated in the respective figure legends. Kmplot was used to compare the survival pro-files of cancer patients with high or low IDH2 expression [24]. The GEO (NCBI) dataset (GDS3101) was utilized to analyze the mRNA levels in the A549 cell line and the corresponding cisplatin-resistant line.

Statistical analysis

Student’s t-test was used to test the statistical difference between two groups of samples, such as cancer tissues compared to normal tissues, or control cells compared to the drug-treated cells. Two-tailed unpaired t-tests were performed using GraphPad Prism 7. A *p* value of less than 0.05 was considered statistically significant.

## 3. Results

### 3.1. High Expression of Wild-Type IDH2 Is Correlated with Poor Survival of Lung Cancer Patients under Chemotherapy and Also Promotes Drug Resistance

Since our previous study showed that wild-type IDH2 was highly expressed in lung cancer and promoted the proliferation of lung cancer cells [23], we analyzed the available data from the Kmplot database to determine the potential correlation between IDH2 expression and the clinical outcome of lung cancer patients. We found that the expression levels of IDH2 were negatively correlated with overall survival (OS) in lung cancer patients treated with chemotherapy (*p* = 0.009), but not in lung cancer patients who did not undergo chemotherapy (Figure 1A,B), suggesting that IDH2 could potentially affect drug sensitivity. Interestingly, the expression of IDH1 was not correlated with patient survival (*p* = 0.3, Figure 1C), indicating that the effect of IDH on drug sensitivity was IDH2 specific. Appendix A shows the relationship between IDH2 expression and survival of lung cancer patients in other subgroups. Male lung cancer patients with high IDH2 expression were at higher risk of poor OS (Appendix A). Higher IDH2 expression was also correlated with worse OS in lung cancer patients undergoing surgical treatment (Appendix A), but it did not significantly affect the survival of lung cancer patients undergoing radiotherapy (*p* = 0.18, Appendix A). Interestingly, the hazard ratio was higher in lung ADC patients than in lung ACC patients (Appendix A). Higher IDH2 expression was also correlated with poorer first progression survival in male lung cancer patients receiving chemotherapy (Appendix A). Together, these data showed that IDH2 could be a prognostic factor for lung cancer patients who received chemotherapy and suggested the possibility that IDH2 might affect the drug sensitivity of lung cancer cells.

Considering that cisplatin is a front-line chemotherapeutic drug of choice for the treatment of lung cancer [7], we analyzed the potential relationship between IDH2 expression and cisplatin sensitivity in lung cancer cells using A549 cell line and its cisplatin-resistant sub-line, A549-CR. Both qRT-PCR and Western blot analyses showed that the cisplatin-resistant A549-CR cells contained a significantly higher level of IDH2 mRNA and protein compared to the parental A549 cells (Figure 1D,E). The higher levels of IDH2 mRNA expression were also consistently observed in the drug-resistant lung cancer cells and ovarian cancer cells in the GEO datasets (Appendix A). Interestingly, the expression levels of lactate dehydrogenase A (LDHA) (Appendix A) and other molecules involved in cisplatin resistance, such as the generation of NADPH and GSH [9], did not change significantly in the cisplatin-resistant cells (Appendix A).

We then tested the impact of IDH2 inhibition on cisplatin response in A549-CR cells. As expected, A549-CR cells were less sensitive to cisplatin than the parental A549 cells (Figure 1F). Inhibition of IDH2 by its pharmacological inhibitor, AGI-6780, significantly increased the sensitivity of A549-CR cells to cisplatin (Figure 1G).

### 3.2. Abrogation of IDH2 Enhances the Sensitivity of Lung Cancer Cells to Cisplatin and Radiation

To further test the role of IDH2 on cisplatin resistance, we generated a stable IDH2-knockdown lung cancer cell line using a lentivirus-mediated shRNA approach. We found that silencing IDH2 expression in A549 cells significantly inhibited colony formation and enhanced cellular sensitivity to cisplatin (Figure 2A,B). Almost all cancer cell colonies were eliminated by shRNA knockdown of IDH2 in combination with 2 μg/mL cisplatin, which by itself (cisplatin) could only inhibit 65% of the colony formation. Similar results were observed in two other assays using MTS assay for cell viability (Figure 2C) and direct counting of cell numbers (Figure 2D). These results could also be seen in another lung cancer cell line HCC827, in which a knockdown of IDH2 by shRNA enhanced the cellular sensitivity to cisplatin in colony formation assay (Figure 2E,F).

We then used AGI-6780, a pharmacological inhibitor of IDH2, to further test its effect on cellular sensitivity to cisplatin and radiation, which is also a major therapeutic modality in lung cancer treatment. Multiple assays, including colony formation, cell viability assays, and direct cell counting, showed that inhibition of IDH2 could significantly enhance the cytotoxic effect of cisplatin in A549 and HCC827 cells (Figure 3A–D). Similarly, AGI-6780 could significantly enhance the effect of radiation on colony formation in lung cancer cells (Figure 3E,F). Together, these data suggest that it would be feasible to use a pharmacological inhibitor of IDH2 to increase the sensitivity of lung cancer cells to conventional chemotherapy and radiotherapy.

### 3.3. Impact of IDH2 on Mitochondrial Metabolism and ROS Generation That Affects Drug Sensitivity

Considering that IDH2 plays an important role in regulating mitochondrial metabolism and redox status [9,18], which could in turn affect cellular response to drug treatment, we further evaluated the relationship between IDH2-mediated metabolic changes and drug sensitivity in lung cancer cells to decipher the underlying mechanisms. We first tested the effect of IDH2 inhibitor AGI-6780 and cisplatin on cellular metabolism using the Seahorse metabolic analyzer to measure the mitochondrial oxygen consumption rate (OCR) and extracellular acidification rate (ECAR). As shown in Figure 4A,B, treatment of A549 cells with 10 μM AGI-6780 and 4 μg/mL cisplatin, the drug concentrations that showed a significant combination effect in cell viability assays, did not have any significant impact on mitochondrial oxygen consumption or glycolytic activity, as there were no substantial changes in OCR or ECAR. We also tested higher drug concentrations and did not observe major changes (Appendix A), except for a slight decrease in basal respiration (Appendix A), maximal respiration (Appendix A), and proton leak (Appendix A) when 10 μM AGI-6780 was combined with 8 μg/mL cisplatin. These results suggested that the drug combination effect on lung cancer cell viability was unlikely to be the result of changes in cellular energy metabolism, which prompted us to test whether IDH2-mediated changes in mitochondrial ROS generation could be a contributing mechanism. 

To determine whether ROS stress might be a potential mechanism for enhancing the cytotoxic effect of cisplatin caused by IDH2 inhibition, we first measured ROS levels in lung cancer cells treated with AGI-6780, cisplatin, or their combination. As shown in Figure 4C, treatment of A549 cells with AGI6780 or cisplatin caused a significant increase in cellular ROS, and combination of the two drugs led to a further increase in ROS, which seemed to correlate with their cytotoxic effect. These changes in ROS were also observed in another lung cancer cell line, HCC827 (Figure 4D). To further test the role of ROS in drug-induced cancer cell death, we pretreated lung cancer cells with the ROS scavenger N-acetyl-cysteine (NAC) and measured its effect on cellular ROS and cell viability. As shown in Figure 4E,F, NAC incubation could significantly decrease intracellular ROS levels in lung cancer cells treated with AGI-6780, cisplatin, or their combination. Importantly, NAC was able to reduce the cytotoxic effect of cisplatin (Figure 4G) or its combination with AGI6780 (Figure 4H).

Two additional assays, colony formation and flow cytometry analysis, were used to confirm the above results. Colony formation assay showed that NAC was able to partially reverse the inhibitory effect of cisplatin in A549 cells whose IDH2 was silenced by shRNA (Figure 5A,B). The protective effect of NAC was also observed in apoptotic analysis using flow cytometry (Figure 5C). Together, these data suggest that ROS generation was likely an important mechanism contributing to the cancer cell death induced by cisplatin and abrogation of IDH2.

### 3.4. Therapeutic Effect of Cisplatin in Mice Bearing Tumor Xenografts with or without IDH2 Knockdown

To evaluate the impact of IDH2 on lung cancer growth in vivo and the therapeutic activity of cisplatin in mice, we performed an animal study using tumor xenografts with A549 cells containing wild-type IDH2 or IDH2 knockdown (stable) via shRNA mediated by a lentiviral vector. The mice were treated with cisplatin (3 mg/kg/week, i.p), and tumor sizes were measured periodically. Consistent with the results of our in vitro experiments, the tumors formed in Balb/C-nude mice implanted with A549 cells containing IDH2-knockdown cells were smaller than those with control cells (Figure 5D, green curve vs. black curve). Cisplatin treatment was able to delay the growth of a tumor containing wild-type IDH2 (red curve vs. black). The inhibitory effect was most potent in the xenografts harboring IDH2-shRNA and treated with cisplatin (blue curve), suggesting that targeting IDH2 could sensitize lung cancer cells to cisplatin in vivo.

## 4. Discussion

Mitochondrial isocitrate dehydrogenase catalyzes the metabolic conversion between isocitrate and α-ketoglutarate, using NAD(P)+/NAD(P)H for the redox reaction in the TCA cycle. As such, IDH2 plays a key role in mitochondrial energy metabolism and redox regulation, which could significantly affect cell survival and proliferation. Our recent study showed that IDH2 enhanced the survival of colon cancer cells by promoting redox homeostasis [15] and leukemia cells by promoting a reductive TCA cycle [16]. In this study, we showed that IDH2 was upregulated in drug-resistant lung cancer cells and enhanced cancer cell survival in the presence of cisplatin treatment. The ability of IDH2 to promote resistance to cisplatin seems to provide an explanation for the poorer clinical outcome in lung cancer patients with high IDH2 expression compared to those with low IDH2 expression among the patients in the chemotherapy group.

Cisplatin exerts its cytotoxic effect through covalent binding with the purine bases in DNA, leading to DNA damage response and activation of signaling pathways that trigger apoptosis [24]. Cancer cells could become resistant to cisplatin through multiple mechanisms, which include an increase in the drug export, upregulation of DNA repair capacity, inactivation of the p53-mediated apoptotic response to DNA damage, and aberrant cell cycle regulation [25]. Based on these mechanisms, various strategies have been developed to overcome cisplatin resistance or sensitize cancer cells to drug treatment. Such strategies include inhibition of glutathione synthesis, suppression of glutathione S-transferases, abrogation of DNA repair, and activation of apoptotic response to DNA damage [24,26,27]. Although these approaches to overcoming cisplatin resistance have met with some success in experimental models and in clinical studies, the overall results are not very satisfactory. Since cisplatin remains as a major front-line chemotherapeutic agent for lung cancer treatment, more effective strategies to overcome cisplatin resistance and enhance its therapeutic activity are needed. Our study showed that IDH2 might play an important role in cisplatin resistance, and that abrogation of IDH2 could increase the sensitivity of lung cancer cells to cisplatin. We used multiple assays to evaluate the effect of the drug on cell viability, cell proliferation, and metabolic activity, with consistent results. However, the effect of IDH2 knockdown on cellular sensitivity to CDDP varied to some degree in multiple assays, although the overall impacts were in the same vein (an increase in drug sensitivity). These data suggest that each assay has its own limitation, and thus the results from multiple assays should be considered together when evaluating the overall impact of IDH2 on cell viability and drug sensitivity.

Interestingly, a recent study showed that treatment of cisplatin-resistant lung cancer cells (H69 and H460) with Riluzole, a drug used in clinical treatment of amyotrophic lateral sclerosis, resulted in an inhibition of lactate dehydrogenase A (LDHA) and xCT antiporter (SLC7A11), leading to a blockage of glutamate transport, an elevation of cellular ROS, and cytotoxicity to lung cancer cells [28]. This study suggests that modulation of cellular redox metabolism and elevated ROS generation could be a potentially effective strategy to overcome cisplatin resistance. Our study, however, showed that LDHA and SLC7A11 were not upregulated in cisplatin-resistant A549 cells (Appendix A). Instead, we found that IDH2 expression was elevated and promoted cancer cell resistance to cisplatin. Importantly, abrogation of IDH2 by shRNA-mediated knockdown or by pharmacological inhibition could significantly sensitize the drug-resistant cells to cisplatin. Induction of ROS stress seems to be a key mechanism for this sensitization effect, since the drug-induced ROS alterations were positively correlated with the cytotoxic effect and antioxidant NAC was able to reverse this sensitization. Although our Seahorse analysis showed no significant changes in OCR when A549 cells were treated with 4–8 μg/mL cisplatin in the presence or absence of AGI-6780, these results do not necessarily suggest that the observed increase in ROS was outside of the mitochondria for the following reasons. The majority of ROS generated in the mitochondria are mainly due to the capture of electrons that “leak” from the respiratory chain by molecular oxygen to form superoxide, which is then converted to H_2_O_2_ by superoxide dismutase (SOD). With this mechanism of ROS generation, it only takes 1–2% of the total oxygen consumption to generate a massive amount of ROS. As such, an alteration in electron transport efficiency due to cisplatin and/or AGI-6780 treatment could cause only a subtle change in OCR, but a significant increase in ROS. Additionally, a decrease in NADPH generation due to IDH2 inhibition by AGI-6780 could also contribute to a ROS increase in the drug combination scenario.

The upregulation of antioxidant capacity often confers drug resistance in cancer cells [8,9]. Indeed, inhibition of antioxidant defense enzymes, and the subsequent disruption of NADPH and GSH production, can sensitize tumors to chemotherapy [13,14]. Our study has identified the mitochondrial metabolic enzyme IDH2 as an important redox regulator in lung cancer cells, in which it helps to maintain redox homeostasis. Inhibition of IDH2 leads to an abnormal accumulation of ROS, likely due to suppression of the IDH2-mediated NADPH generation, and renders the cancer cells more vulnerable to ROS stress induced by cisplatin. Furthermore, since the upregulation of antioxidant capacity is an important mechanism for radiation tolerance and targeting mitochondrial metabolism is considered as a potential strategy to reverse radio resistance [29], the unique roles of IDH2 in both redox regulation and mitochondrial metabolism seem to make this enzyme a potentially important target for sensitization to radiotherapy. Our study showed that inhibition of IDH2 indeed increased the sensitivity of lung cancer cells to radiation in colony formation assay. Further in vitro and in vivo studies are needed to test if IDH2 could be targeted to enhance the efficacy of radiotherapy.

It is of particular interest to note that the pro-tumor role of wild-type IDH2 and its underlying mechanisms are different from the oncogenic function of IDH2 mutations, which are most often observed in certain gliomas and leukemia, such as acute myeloid leukemia (AML) [30,31]. Mutation in IDH2 leads to increased generation of oncogenic metabolite 2-hydroxyglutarate (2-HG), which promotes carcinogenesis by causing alterations in epigenetics, whereas wild-type IDH2 promotes cancer cell survival and proliferation through enhancing the reductive TCA cycle for the utilization of glutamine for lipid synthesis and increasing the antioxidant capacity of the cancer cells to counteract oxidative stress [15]. It is important to note that the clinical drug Enasidenib (AG-221) used in clinical treatment of AML is a specific inhibitor of mutant IDH2 [17], and thus is unlikely to inhibit wild-type IDH2 effectively. Our study showed that a specific knockdown of wild-type IDH2 in lung cancer cells increased their sensitivity to cisplatin in a mouse xenograft model. It would be important to test if the IDH2 inhibitor AGI-6780 could also potentiate antitumor activity of cisplatin in animals. To enhance the therapeutic effect of cisplatin or radiation via inhibition of IDH2, new inhibitors of wild-type IDH2 still need to be developed and evaluated in experimental models and in clinical settings.

## 5. Conclusions

We show that IDH2 is elevated in cisplatin-resistant lung cancer cells and promotes tumor cell survival through a mitochondrial redox metabolism to enhance antioxidant capacity. Inhibition of IDH2 increases the sensitivity of lung cancer cells to cisplatin, and potentially radiation, through a ROS-mediated mechanism. Our study suggests that mitochondrial wild-type IDH2 could be a potential new target to enhance the therapeutic effect of cisplatin and radiotherapy in lung cancer.

## Figures and Tables

**Figure 1 biomedicines-11-00475-f001:**
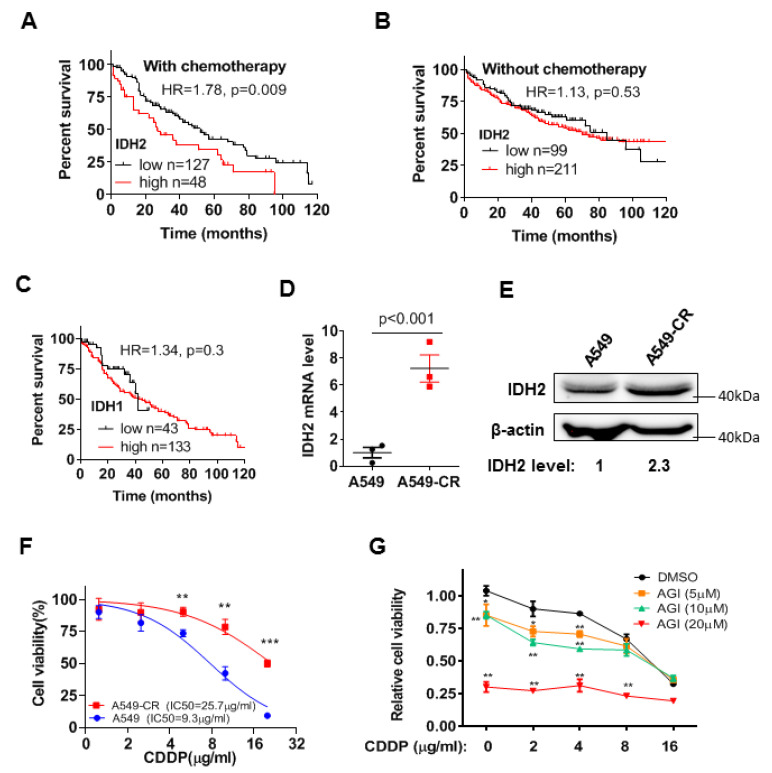
Impact of IDH2 expression on lung cancer patient survival and cellular sensitivity to cisplatin. (**A**,**B**) Kaplan–Meier survival curves of lung cancer patients treated with chemotherapy (**A**) or without chemotherapy (**B**), stratified by IDH2 expression levels. Datasets were from Kmplot lung database, as described in Methods. The Kaplan Meyer curves analysis was performed using the software tool provided in the Kaplan–Meier Plotter website. The patients were divided using auto select best cutoff method; the subtype was restricted to lung cancer patients who were treated with or without chemotherapy and whose survival data were available. The datasets were integrated into one large group for analysis using the method/tool provided on Kaplan–Meier Plotter website. The patients with survival time of less than 120 months (10 years, with whole number in months) were used for analysis of Hazard Ratio and *p* value using Log-rank (Mantel-Cox) test (GraphPad prism software). (**C**) Kaplan–Meier survival curves of lung cancer patients with chemotherapy, stratified by IDH1 expression levels. The best-performing threshold value of IDH1 expression was used as the cut-off. (**D**,**E**) Relative mRNA levels of IDH2 in A549 parental and cisplatin-resistant cells, measured via qRT-PCR and Western blot analyses. (**F**) Effect of cisplatin on cell proliferation in A549 cells and the cisplatin-resistant A549-CR cells. The lung cancer cells were incubated with the indicated concentrations of cisplatin for 48 h, and cell viability was measured via MTS assay. (**G**) Impact of the IDH2 inhibitor AGI6780 on the cytotoxic effect of cisplatin. A549-CR cells were incubated with the indicated concentrations of cisplatin and AGI6780 for 48 h, and cell viability was measured via MTS assay. * *p* ≤ 0.05; ** *p* ≤ 0.01; *** *p* ≤ 0.001.

**Figure 2 biomedicines-11-00475-f002:**
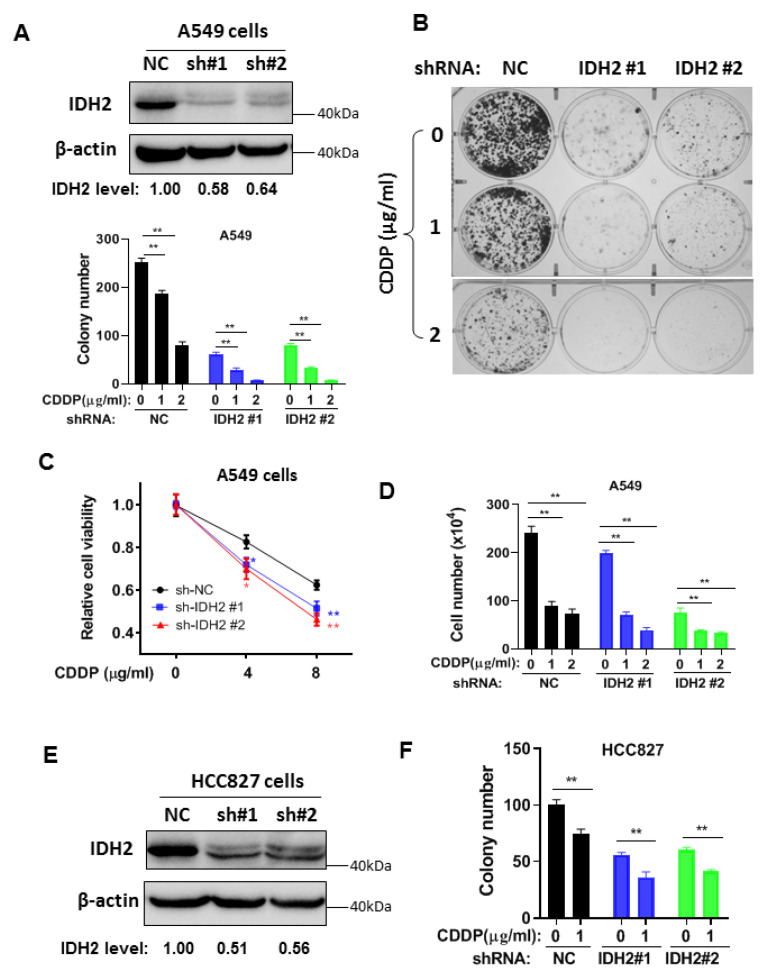
Silencing of IDH2 by shRNA enhanced the sensitivity of lung cancer cells to cisplatin. (**A**,**B**) Colony formation of A549 cells stably transfected with IDH2 shRNA (IDH2 #1 and #2) or non-targeting control shRNA (NC). The knockdown of IDH2 protein was confirmed via Western blot analysis (upper right panel); colonies were counted on day 14. (**C**,**D**) A549 cells were stably transfected with IDH2 shRNA (sh-IDH2 #1 and #2) or non-targeting control shRNA (sh-NC). Cell viability (**C**) and cell (**D**) were measured 48 h after drug incubation. (**E**,**F**) Western blot (**E**) and colony number (**F**) of HCC827 cells stably transfected with IDH2 shRNA (#1 and #2) or control shRNA (NC). Colonies were counted on day 14. * *p* ≤ 0.05; ** *p* ≤ 0.01.

**Figure 3 biomedicines-11-00475-f003:**
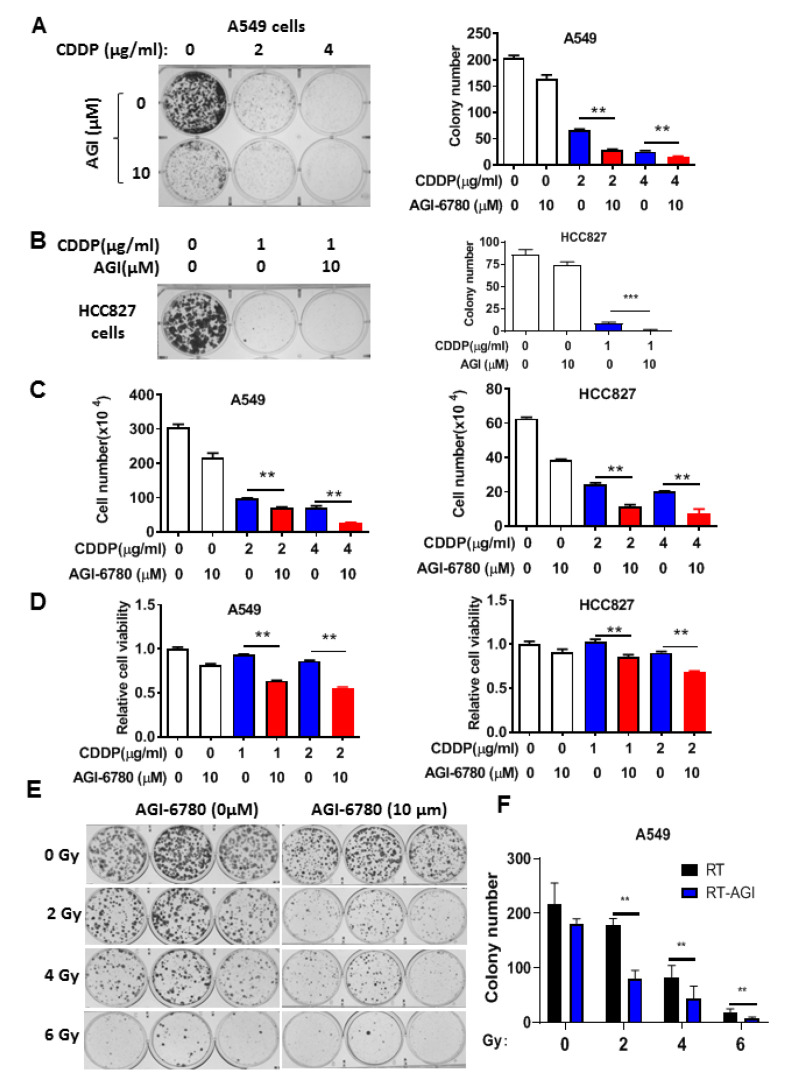
Pharmacological inhibition of IDH2 rendered lung cancer cells more sensitive to cisplatin and radiation. (**A**,**B**) Colony formation of A549 (**A**) or HCC827 (**B**) cells incubated with the indicated concentrations of cisplatin and/or AGI-6780. Colonies were counted on day 14. (**C**) A549 and HCC827 cells incubated with the indicated concentrations of cisplatin or/and AGI-6780; cell numbers were measured 48 h after drug incubation. (**D**) A549 and HCC827 cells were incubated with the indicated concentrations of cisplatin or/and AGI-6780. Cell viability was measured via MTS assay 48 h after drug incubation. (**E**,**F**) A549 cells were treated with the indicated doses of radiation in the presence or absence of 10 μM AGI-6780. Colonies were counted on day 14. ** *p* ≤ 0.01; *** *p* ≤ 0.001.

**Figure 4 biomedicines-11-00475-f004:**
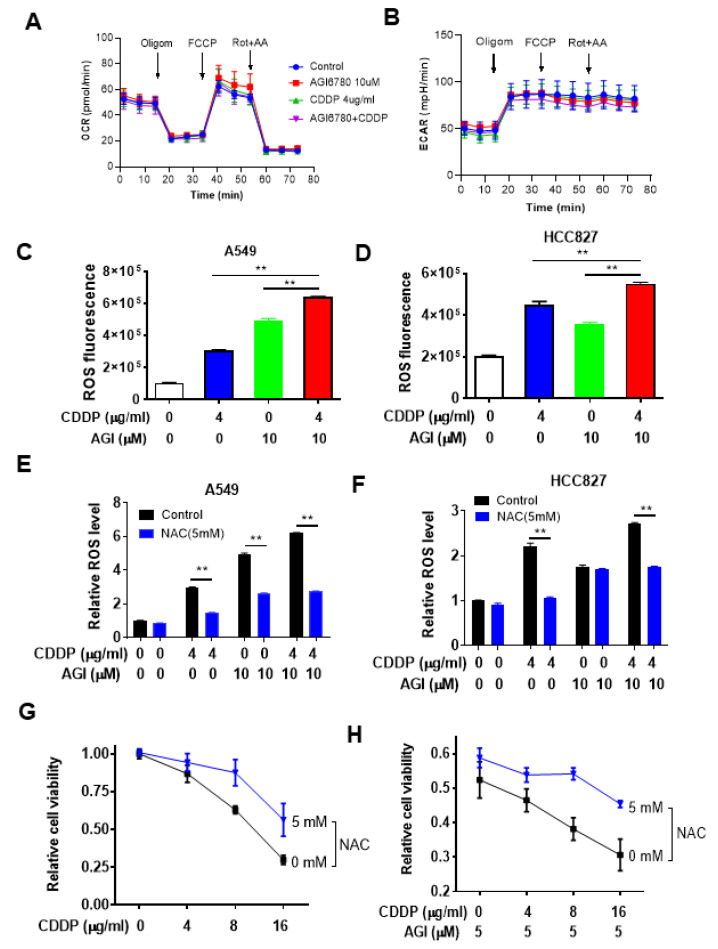
Impact of IDH2 abrogation on oxygen consumption and ROS and their effect on the sensitivity of lung cancer cells to cisplatin. (**A**,**B**) Measurement of cellular oxygen consumption rate (OCR) (**A**) and extracellular acidification rate (ECAR) (**B**) in A549 cells treated with AGI6780 and CDDP (4 μg/mL), as indicated. OCR and ECAR were measured using a Seahorse XF24 metabolic analyzer. (**C**,**D**) A549 (**C**) or HCC827 (**D**) cells were treated with cisplatin or/and AGI-6780, as indicated, for 48 h. Intracellular ROS levels were measured by staining cells with dihydroethidium, followed by detection of fluorescence using flow cytometer analysis. (**E**,**F**) A549 (**E**) or HCC827 (**F**) cells were pretreated with N-Acetyl-l-Cysteine (NAC) for 1 h and then incubated with the indicated concentrations of cisplatin and/or AGI-6780 for 48 h. Cellular ROS levels were measured by staining cells with dihydroethidium, followed by detection of fluorescence using flow cytometer analysis. (**G**,**H**) A549 cells were pretreated with 5 mM NAC for 1 h, then incubated with the indicated concentrations of cisplatin alone (**G**) or with cisplatin + AGI-6780 combination (**H**). Cell viability was measured 48 h after drug incubation. ** *p* ≤ 0.01.

**Figure 5 biomedicines-11-00475-f005:**
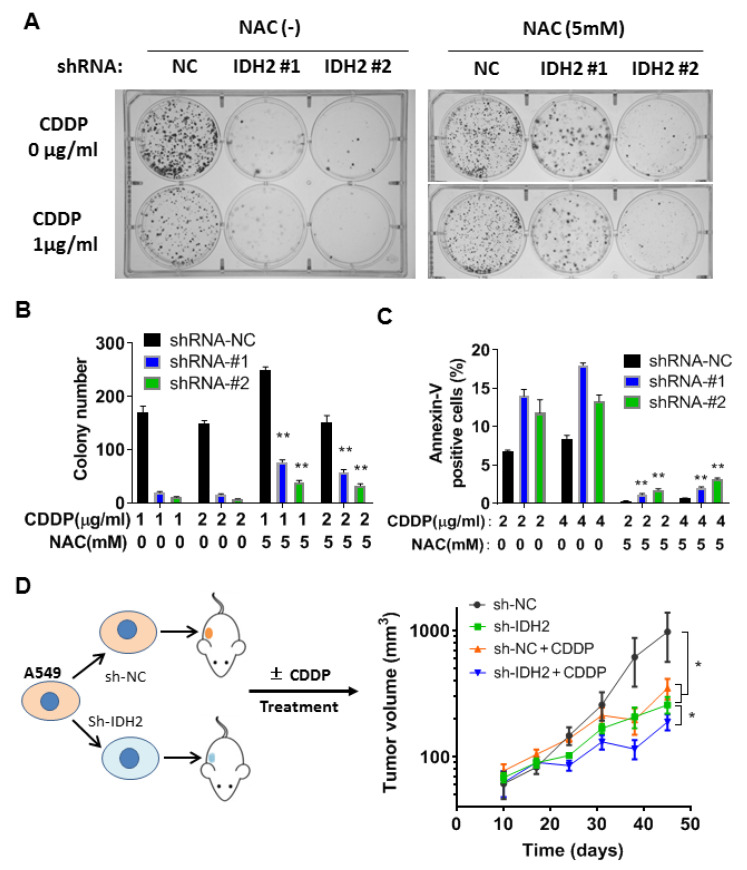
Combination of IDH2 abrogation and cisplatin exhibited potent anticancer activity in vitro and in vivo. (**A**,**B**) A549 stably transfected with IDH2 shRNA (#1 and #2) or control shRNA (NC) were pretreated with or without NAC and incubated with cisplatin (CDDP), as indicated. Colonies were counted on day 14. (**C**) A549 cells were incubated with the indicated concentrations of CDDP and/or NAC, and cell death was quantified using Annexin V-FITC/PI staining followed by flow cytometry analysis. (**D**) Balb/C-nude mice were implanted with A549 cells with stable transfection with IDH2 shRNA (sh-IDH2 #1) or control shRNA (sh-NC), as illustrated in the left panel. Mice were treated without or with cisplatin (3 mg/kg/week, i.p.). Tumor sizes were measured at the indicated time points. *, *p* ≤ 0.05; **, *p* ≤ 0.01.

## Data Availability

The datasets used and analyzed during the current study are available from the corresponding author on reasonable request. The data used for survival analysis of lung cancer patients were downloaded from the public database Kaplan–Meier Plotter (http://kmplot.com/analysis/index.php?p=background, accessed on 1 November 2020). The data used for mRNA analysis of cisplatin resistant cells were from GEO database (https://www.ncbi.nlm.nih.gov/gds/?term=GDS3101, accessed on 1 November 2020).

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
