# Peer review of "Targeting Mitochondrial IDH2 Enhances Antitumor Activity of Cisplatin in Lung Cancer via ROS-Mediated Mechanism"

_biomedicines, 2023, doi:10.3390/biomedicines11020475_

Round 1
Reviewer 1 Report
In this paper, the authors showed that Inhibition of IDH2 increases the sensitivity of lung cancer cells to cisplatin through ROS-mediated mechanism. I read the study with interest. The data are novel and informative.
In addition, the manuscript is well written and this is a potentially interesting manuscript.
I do not have any substantial amendments to suggest. I have a few relatively comments, explained below.
Minor
1. In conclusion, the authors insisted that Inhibition of IDH2 increases the sensitivity of lung cancer cells to cisplatin and “radiation” through ROS-mediated mechanism. However, the experiments of radiation were only Figure 3 E, F. The conclusion should be toned down to reflect the data.
2. It would be helpful if the authors would explain the assay used to measure MTS assay in more details.
3. Details of how radiation was performed is lacking in the Methods section.
4. How many times was the experiment repeated?
5. In western blots, molecular size markers should be indicated on each blot.
6. In method, the anti-IDH2 antibody should be more detailed.
Author Response
The authors would like to thank the review for the constructive comments and helpful suggestions, which are highly valuable for us to strengthen our manuscript.
- “In conclusion, the authors insisted that Inhibition of IDH2 increases the sensitivity of lung cancer cells to cisplatin and “radiation” through ROS-mediated mechanism. However, the experiments of radiation were only Figure 3 E, F. The conclusion should be toned down to reflect the data.”
Response:Thank you for this suggestion. We have now toned down the conclusion on radiation as suggested. Now it reads as “Inhibition of IDH2 increases the sensitivity of lung cancer cells to cisplatin and potentially radiation through ROS-mediated mechanism.” (page 14 line 445 revisions mode)
- “It would be helpful if the authors would explain the assay used to measure MTS assay in more details.”
Response:We have included a description of MTS assay under Materials and Methods section as suggested (page 3 line 106-109 revisions mode).
- “Details of how radiation was performed is lacking in the Methods section.”
Response:The cell radiation experiment is now described as a separate section (lines 115-119).
- “How many times was the experiment repeated?”
Response:The experiments were performed in triplicates. For key experiments such as comparison of drug sensitive and resistant A549 cells and CDDP + AGI drug combination experiments, the triplicate experiments were independently repeats twice.
- “In western blots, molecular size markers should be indicated on each blot.”
Response: We have new included molecular size markers on each western blot as suggested.
- “In method, the anti-IDH2 antibody should be more detailed.”
Response: the anti-IDH2 antibody was from Abcam (ab55271). This information has been added in the Methods section (line 84 revisions mode).
Reviewer 2 Report
Li et al. reported that IDH2 genetic/pharmacological inhibition could increase the sensitivity of lung cancers cells to cisplatin. However, some issues deriving from the experimental model/conditions affect the presented data.
Major concerns:
1) For Kaplan Meyer curves presented in Fig1 and Supp Figure1 the author used Kaplan-Meier Plotter database. To allow the reproducibility of the analysis the authors should report details of this analysis such as methods to split patients in high and low category, subtypes analyzed, treatment groups, datasets used and numbers of patients in each reported in each plot
2) The cellular model of cisplatin resistant cells (A549-CR) used in the manuscript presens some big issues:
i) A549-CR are cultured in My-co5A with 10% FBS. Is My-co5A a typo for McCoy’s 5A media? If yes, why the authors culture A549-CR in McCoy’s 5A while A549 in DMEM? When compared, did the authors use the same media? The authors should clarify this point in the methods section. Moreover, details about the development of this resistant variants are lacking (e.g. number and doses of cisplatin treatment)
ii) the “parental” a549 used were comparable with A549-CR in terms of passages? Did they derive from the same A549 developed in the same lab? This is of pivotal importance because continuous cell culture expansion (i.e. time to derive a resistant cell line) could lead to the acquisition of modification. The use of a parental cell line that undergoes the same exact manipulation except for cisplatin treatment is the gold standard in literature.
iii) In Figure 1F, A549-CR were not more resistant in comparison to parental ones as stated by the authors (line 197). Indeed, no statistical test was carried out and levels of IC50 seems to be equal. Since A549-CR is the main model used in this manuscript, this problem affects the overall conclusion drawn from the authors.
3) In figure 1 G, the authors did not carry out any statistical test and analysis of typical drug sensitivity metric (i.e. IC50, AUC). Moreover, AGI treatment at basal levels (w/o cisplatin) already strongly affect cell viability, thus is not clear whether the effect is depending from the inhibition of IDH2 or from the fact the AGI was used at very high “toxic” doses. The author should perform additional experiment to clarify this point. More importantly, as reported in the point 2, A549-CR are not cisplatin-resistant and therefore the authors are not answering to their initial hypothesis.
4) The figure 2A should be reformatted for reader clarity. The western blot should come first (Figure 2 A) and the colony formation assay with the relative quantification should become Figure 2B. Moreover, why the authors choose the dose of 1 and 2 uM to perform the experiment? How dose cisplatin was dissolved and how long the cells were treated? To better appreciate the effects observed in Figure 2A, colony number should be presented also as fold change for each cell lines relative to not treated (CDDP 0uM). This will allow the reader to really understand the effect of IDH2 silencing in the modulation of cisplatin resistance.
5) Data obtained in Figure 2D (cell number) seems to be in contrast to the one obtained with MTT. Why? Moreover, why the authors did not use the same drug concentration? I would recommend to use the same experimental conditions.
6) Why did the author choose HCC827 cell line to confirm data in other cellular models? HCC827 is an EGFR mutated cell lines and it is not a proper model to assess cisplatin sensitivity since in clinics EGFR mutated patients are treated with EGFR TKI inhibitors. I would recommend data validation in other cells with no druggable mutation in clinics.
7) Is cisplatin and AGI combination effect synergic or additive?
8) Figure 4. The authors showed the effect of AGI and cisplatin treatment on metabolism. However, data obtained with AGI should be validated using a genetic model (i.e. A549 cell line with sh-IDH2) to exclude any aspecific effects of the drug. By looking at the previous publication from the authors, it seems that IDH2 modulation affected OCR. Why AGI treatment not? Is the dose used inhibiting IDH2?
9) Figure 5. Since authors are proposing IDH2 inhibition in combination with cisplatin, in vivo experiment using AGI treatment are warranted.
Minor Concern
1) Put quantification in each blot in order to homogeneize all the figures presented in the manuscript.
2) At line 43-44, the author stated that cancer, platinum-based conventional chemotherapies remain as the most commonly used effective treatment [3-5]. With the advent of immunotherapy, this scenario is rapidly changing. Please revise this sentence accordingly.
3) Which antibodies was used for western blot analysis? Please cite it in Methods section.
Author Response
The authors would like to thank the review for the constructive comments and helpful suggestions, which are highly valuable for us to strengthen our manuscript.
1) “For Kaplan Meyer curves presented in Fig1 and Supp Figure1 the author used Kaplan-Meier Plotter database. To allow the reproducibility of the analysis the authors should report details of this analysis such as methods to split patients in high and low category, subtypes analyzed, treatment groups, datasets used and numbers of patients in each reported in each plot.”
Response: The Kaplan Meyer curves analysis was performed using the software tool provided in the Kaplan-Meier Plotter website. The patients were divided using auto select best cutoff method; the subtype was restricted to lung cancer patients who were treated with or without chemotherapy and whose survival data were available. The datasets were integrated into one large group for analysis using the method/tool provided in Kaplan-Meier Plotter website. The patients with survival time of less than 120 months (10 years, with whole number in months) were used for analysis of Hazard Ratio and P value using Log-rank (Mantel-Cox) test (GraphPad prism software). This information is now added into figure legend now lines 219-226 revisions mode). The numbers of patients were now indicated in the respected figures.
2) “The cellular model of cisplatin resistant cells (A549-CR) used in the manuscript presens some big issues: i) A549-CR are cultured in My-co5A with 10% FBS. Is My-co5A a typo for McCoy’s 5A media? If yes, why the authors culture A549-CR in McCoy’s 5A while A549 in DMEM? When compared, did the authors use the same media? The authors should clarify this point in the methods section. Moreover, details about the development of this resistant variants are lacking (e.g. number and doses of cisplatin treatment). ii) the “parental” a549 used were comparable with A549-CR in terms of passages? Did they derive from the same A549 developed in the same lab? This is of pivotal importance because continuous cell culture expansion (i.e. time to derive a resistant cell line) could lead to the acquisition of modification. The use of a parental cell line that undergoes the same exact manipulation except for cisplatin treatment is the gold standard in literature. iii) In Figure 1F, A549-CR were not more resistant in comparison to parental ones as stated by the authors (line 197). Indeed, no statistical test was carried out and levels of IC50 seems to be equal. Since A549-CR is the main model used in this manuscript, this problem affects the overall conclusion drawn from the authors.”
Response: Thank you for these insightful comments on cell culture conditions and cell passages, which could indeed affect cellular sensitivity to drugs. In our original experiments, A549 and A549-CR cells were cultured in DMEM and McCoy’s 5A media, respectively, as recommended by the vendors. We agree with the reviewer that cell culture conditions could significantly affect drug sensitivity, and thus re-performed the experiments on CDDP cytotoxicity using the same medium. A549 and A549-CR cells were cultured in DMEM+10% FBS for several days before subjected to CDDP cytotoxicity experiments, which were repeated twice. The new results showed a significant difference in CDDP sensitivity of the two cell lines. We have now replaced the previous Figure 1F with these new results. Statistical analysis was also performed with standard deviation bars and p values indicated in the new figure 1F. We have also noticed that the CDDP concentration unit was mistakenly labeled as “uM”, which should be “ug/ml” in our records. This labeling error has been corrected in all figures. Since A549-CR cell line was purchased from Cell Resource Center, Peking Union Medical College (Beijing, China), the cells were not at the same number of passages with the A549 cells maintained in our lab. While we agree that cell passages could affect drug sensitivity, it might not be a significant issue in our study because the main purpose was to exam the relationship between IDH2 expression and CDDP sensitivity. A549-CR cells showed higher level of IDH2 expression and were more resistant to CDDP than A549 cells, consistent with the notion that IDH2 might contribute to CDDP resistance.
3) “In figure 1 G, the authors did not carry out any statistical test and analysis of typical drug sensitivity metric (i.e. IC50, AUC). Moreover, AGI treatment at basal levels (w/o cisplatin) already strongly affect cell viability, thus is not clear whether the effect is depending from the inhibition of IDH2 or from the fact the AGI was used at very high “toxic” doses. The author should perform additional experiment to clarify this point. More importantly, as reported in the point 2, A549-CR are not cisplatin-resistant and therefore the authors are not answering to their initial hypothesis.”
Response: We used multiple concentrations of AGI and CDDP in Figure 1G, which showed that at the relatively low drug concentrations (AGI 5-10 uM, CDDP 2-4 ug/ml), combination of the two drugs was more effective. However, at the high drug concentrations (AGI 20 uM, CDDP 8-16 ug/ml), such potentiation effect disappeared due to highly toxic effect of each drug at high concentrations. Regarding A549-CR cells, the new results showed that they are more resistant to CDDP than A549 cells when both cell line were cultured in the same medium (see point 1 above).
4) “The figure 2A should be reformatted for reader clarity. The western blot should come first (Figure 2 A) and the colony formation assay with the relative quantification should become Figure 2B. Moreover, why the authors choose the dose of 1 and 2 uM to perform the experiment? How dose cisplatin was dissolved and how long the cells were treated? To better appreciate the effects observed in Figure 2A, colony number should be presented also as fold change for each cell lines relative to not treated (CDDP 0uM). This will allow the reader to really understand the effect of IDH2 silencing in the modulation of cisplatin resistance.”
Response: We have inverted the order of Figure 2A and 2B as suggested, and the experimental conditions were described in more detail on lines 110-111 (revisions mode). Regarding the bar graph of colony formation in Fig 2B, we tried to use fold change for each cell lines relative to not treated (CDDP 0 uM) as suggested (see the left panel in the figure below), but found that this type of data conversion did now clearly show the combined effect of IDH2 knockdown and CDDP treatment on colony formation. Thus, we would still prefer to present the data as actual colony numbers (right panel) in Figure 2B.
5) “Data obtained in Figure 2D (cell number) seems to be in contrast to the one obtained with MTT. Why? Moreover, why the authors did not use the same drug concentration? I would recommend to use the same experimental conditions.”
Response: This is a good point. It seemed that counting the actual cell numbers showed more impact of CDDP on the cell proliferation compared to the MTS assay, which mainly measures the dehydrogenase activity in the cells. Thus, it is better is use multiple assays (as in our study) to evaluate the drug effect on cell proliferation and metabolic activity. In our study, the effect of IDH2 knockdown on cellular sensitivity to CDDP was consistent (same direction) in multiple assays, although the level of impact did vary to some degree. We agree that it would be better to use the same drug concentrations in all experiments. Due to time constrain (10 days for revision), we did not repeat all experiments using same drug concentrations.
6) “Why did the author choose HCC827 cell line to confirm data in other cellular models? HCC827 is an EGFR mutated cell lines and it is not a proper model to assess cisplatin sensitivity since in clinics EGFR mutated patients are treated with EGFR TKI inhibitors. I would recommend data validation in other cells with no druggable mutation in clinics.”
Response: The survival analysis of chemotherapy show IDH2 expression was more significantly affect survival of lung adenocarcinoma compared with squamous cell lung carcinoma (supplement Fig. S1D, E), so we choose lung adenocarcinoma cell lines to study. Although HCC827 is EGFR mutated cell lines, it is more sensitive to CDDP treatment (Fig. 3 A, B) and also show increase ROS level with AGI and CDDP combination (Fig. 4 C, D). This seems to suggest that EGFR mutated cells could be sensitive to CDDP treatment, at least in vitro.
7) “Is cisplatin and AGI combination effect synergic or additive?”
Response: The effect of cisplatin and AGI combination appeared additive (Fig. 2 B, C; Fig. 4 C, D).
8) “Figure 4. The authors showed the effect of AGI and cisplatin treatment on metabolism. However, data obtained with AGI should be validated using a genetic model (i.e. A549 cell line with sh-IDH2) to exclude any specific effects of the drug. By looking at the previous publication from the authors, it seems that IDH2 modulation affected OCR. Why AGI treatment not? Is the dose used inhibiting IDH2?”
Response: Although sh-IDH2 cells are a stable genetic model, the long-time selection process might provide a sufficient time for the cells compensate the metabolic disturbance due to IDH2 knockdown. Thus, chemical inhibitor AGI was used in this study to test the acute effect of IDH2 inhibition on OCR and ROS metabolism. In A549 cells, AGI at the concentration (10 uM) that enhanced the cellular sensitivity to CDDP mainly promoted ROS generation without a significant effect on OCR. This was different from leukemia cells, which are highly sensitive to AGI inhibition, likely due to their high dependency on IDH2 for TCA cycle (PMID: 35313945).
9) “Figure 5. Since authors are proposing IDH2 inhibition in combination with cisplatin, in vivo experiment using AGI treatment are warranted.”
Response: This point is well taken. The animal experiment in Figure 5D aimed to provide in vivo evidence that targeting IDH2 could enhance the therapeutic activity of CDDP. Due to the limited time provided for revision (10 days), we did not perform animal experiments using AGI as suggested.
Minor Concern
1) “Put quantification in each blot in order to homogeneize all the figures presented in the manuscript.”
Response: We have added quantification numbers as suggested.
2) “At line 43-44, the author stated that cancer, platinum-based conventional chemotherapies remain as the most commonly used effective treatment [3-5]. With the advent of immunotherapy, this scenario is rapidly changing. Please revise this sentence accordingly.”
Response: The sentence is changed as“With the advance in cancer treatment including immunotherapy, platinum-based chemotherapy still remain as a commonly used treatment option.”
3) “Which antibodies was used for western blot analysis? Please cite it in Methods section.”
Response: The anti-IDH2 antibody was from Abcam (ab55271) and β-actin was from Cell Signaling Technology (Danvers, MA, USA, cst4970). This information has been added in the methods (Lines 84-85, revisions mode).
Reviewer 3 Report
Biomedicines-2143502
The article, titled: 'Targeting mitochondrial IDH2 enhances antitumor activity of cisplatin in lung cancer via ROS-mediated mechanism', is interesting and raises important issues related to resistance to cisplatin treatment.
Below, are some comments on the manuscript.
1. Throughout the article, in vitro and in vivo should be corrected so that the notation is in italics.
2. There are a lot of abbreviations in the article. The first time a particular abbreviation is used, its full name should be given.
3. Line 41: '...squamous cell lung carcinoma (ACC)...; is the correct abbreviation in parentheses?
4. Line 72-73: ‘We also examined the effects of combining IDH2 inhibition with cisplatin or radiation for the treatment of lung cancer… Is it cell line studies? If so, this should be noted.
5. Line 114: ‘…contained the indicated cDNA, primers, and SYBR®Green Real-Time PCR Master Mixes…’ It would be advisable to indicate to which regions the primers were complementary and provide the sequences of the primers.
6. Line 170: ‘…in lung cancer patients who did not undergo chemotherapy…’ For what reason were they not given chemotherapy? Was it due to their generally poor condition and inability to receive chemotherapy, or was it due to the presence of predictive factors qualifying them for other therapies? This should be clarified. It should also be mentioned here that there is no basic clinical-demographic data in the article regarding the study group - the patients' characteristics. this should be supplemented.
7. The authors did some of their research in vitro, some in vivo, and some in silico. It is sometimes possible to get lost in the article in this regard. It would need to be cleaned up a bit, e.g. indicating when discussing results obtained by analyzing available data from databases that these were in silico studies, etc.
8. Line 197: ‘As expected, A549-CR cells were less sensitive to cisplatin than the parental A549 cells.’ Is this sentence even necessary? If the cell line is of the cisplatin resistance type, it is unlikely to be any breakthrough discovery that cisplatin resistance was found in it.
9. Figures 1 A and 1B lack an indication that this is an IDH2 analysis. This should be inserted either in the header or when describing the curves (low, high) of K-M.
Author Response
The authors would like to thank the review for the constructive comments and helpful suggestions, which are highly valuable for us to strengthen our manuscript.
- “Throughout the article, in vitro and in vivo should be corrected so that the notation is in italics.”
Response: The words “in vitro” and “in vivo” are now italicized as suggested.
- “There are a lot of abbreviations in the article. The first time a particular abbreviation is used, its full name should be given.”
Response: The full names of the abbreviations are now provided when they appear for the first time in the text.
- “Line 41: '...squamous cell lung carcinoma (ACC)...; is the correct abbreviation in parentheses?”
Response: The mistake (ACC) is now corrected (SCC).”
- “Line 72-73: ‘We also examined the effects of combining IDH2 inhibition with cisplatin or radiation for the treatment of lung cancer… Is it cell line studies? If so, this should be noted.”
Response: The effects of combining IDH2 inhibition with radiation for suppression of lung cancer cell colony formation were studied and the results are shown in Fig. 3E, F.
- “Line 114: ‘…contained the indicated cDNA, primers, and SYBR®Green Real-Time PCR Master Mixes…’ It would be advisable to indicate to which regions the primers were complementary and provide the sequences of the primers.”
Response: The sequences of the primers for IDH2 and beta-actin are now provided in the Method section under RT-qPCR.
- “Line 170: ‘…in lung cancer patients who did not undergo chemotherapy…’ For what reason were they not given chemotherapy? Was it due to their generally poor condition and inability to receive chemotherapy, or was it due to the presence of predictive factors qualifying them for other therapies? This should be clarified. It should also be mentioned here that there is no basic clinical-demographic data in the article regarding the study group - the patients' characteristics. this should be supplemented.”
Response: The information of lung cancer patients who undergo chemotherapy or did not undergo chemotherapy was downloaded from the database of Kmplot (http://kmplot.com/analysis/index.php?p=background). Sources for the databases include GEO, EGA, and TCGA. Thus, the reasons for different treatment options and patients clinical characteristics are unavailable to us.
- “The authors did some of their research in vitro, some in vivo, and some in silico. It is sometimes possible to get lost in the article in this regard. It would need to be cleaned up a bit, e.g. indicating when discussing results obtained by analyzing available data from databases that these were in silico studies, etc.”
Response: The statement of available data from databases has been added in abstract (line 17) and in Result 3.1 (line 179-180) to indicate the data types. Results 3.2- 3.4 are all experimental data for cell or animal study.
- “Line 197: ‘As expected, A549-CR cells were less sensitive to cisplatin than the parental A549 cells.’ Is this sentence even necessary? If the cell line is of the cisplatin resistance type, it is unlikely to be any breakthrough discovery that cisplatin resistance was found in it.”
Response: This sentence and Figure 1F indicate that A549-CR is a cell line resistant to cisplatin. This is necessary to provide the background information for A549-CR cells.
- “Figures 1 A and 1B lack an indication that this is an IDH2 analysis. This should be inserted either in the header or when describing the curves (low, high) of K-M.”
Response: “IDH2” is now added in Figures 1A and 1B.
Round 2
Reviewer 2 Report
In the revised form of the manuscript, the authors replied to the majority of the issues raised.
Few points were not further explored due to time constraints of the revision. While this is clearly conceivable, I would suggest to briefly discuss these limitations in the discussion section.
Minor points:
1. Figure 1 G is still lacking statistics
2. Typo at line 110 (warmed warter)
Author Response
Point 1 “In the revised form of the manuscript, the authors replied to the majority of the issues raised. Few points were not further explored due to time constraints of the revision. While this is clearly conceivable, I would suggest to briefly discuss these limitations in the discussion section.”
Response: Thank you for these comments. As suggested, we have now included a discussion on potential limitations in some of our assays and in vivo study (lines 391-399 and lines 450-453).
Minor points:
“1. Figure 1 G is still lacking statistics; 2. Typo at line 110 (warmed warter)”
Response: We have now included statistical p values in Figure 1G, and have also corrected the typo in line 111. We have also provided p values in other figures.
Reviewer 3 Report
The efforts made by the authors to make corrections to the manuscript should be appreciated. The only note is that the changes made were not marked (e.g. in the 'track changes' mode, or marked in color). But most importantly, they were applied.
Author Response
Point 1 “The efforts made by the authors to make corrections to the manuscript should be appreciated. The only note is that the changes made were not marked (e.g. in the 'track changes' mode, or marked in color). But most importantly, they were applied.
Response: The word file with “track changes” was submitted with a “clean version”. It seems that the clean version was sent to the reviewers. We apologize for this confusion.